

# Short communication: Multiscalar drag decomposition in fluvial systems using a transform-roughness correlation (TRC) approach

David L. Adams[1,2] and Andrea Zampiron[3]

[1]Department of Geography, The University of British Columbia, Vancouver, BC, Canada
[2]School of Geography, The University of Melbourne, Melbourne, VIC, Australia
[3]Department of Mechanical Engineering, The University of Melbourne, Melbourne, VIC, Australia

**Correspondence:** David L. Adams (dladams@alumni.ubc.ca)

**Abstract.**

In natural open-channel flows over complex surfaces, a wide range of superimposed roughness elements may contribute to flow resistance. Gravel-bed rivers present a particularly interesting example of this kind of multiscalar flow resistance problem, as both individual grains and bedforms can potentially be important roughness elements. In this paper, we propose a novel

method of estimating the relative contribution of different physical scales of river bed topography to the total drag, using a transform-roughness correlation (TRC) approach. The technique, which requires only a single longitudinal profile, consists of (1) a wavelet transform which decomposes the surface into roughness elements occurring at different wavelengths, and (2) a 'roughness correlation' that estimates the drag associated with each wavelength based on its geometry alone, expressed as $k_s$. We apply the TRC approach to original and published laboratory experiments and show that the multiscalar drag decomposition

yields estimates of grain- and form-drag that are consistent with estimates in channels with similar morphologies. Also, we demonstrate that the roughness correlation may be used to estimate total flow resistance via a conventional equation, suggesting that it could replace representative roughness values such as median grain size or the standard deviation of elevations. An improved understanding of how various scales contribute to total flow resistance may lead to advances in hydraulics as well as channel morphodynamics.

## 1  Introduction

Understanding flow resistance is of great interest to river research and practice. The estimation of flow resistance is important for determining flood magnitudes, predicting ecological habitat, and understanding the morphodynamic behaviour of channels. However, the hydraulics of gravel-bed channels, in particular, are relatively poorly understood due to a range of factors (see Ferguson, 2007). Given that most of the foundational work in fluid dynamics, upon which conventional approaches to predict-

ing flow resistance are based, was conducted using regular (e.g., Schlichting, 1936) or uniscalar (e.g., Nikuradse, 1933) bed geometry, the multiscalar topographic characteristics of these rivers presents a major challenge. In particular, both individual grains and bedforms on the bed surface, spanning orders-of-magnitude of scale, may contribute to the total flow resistance (see Adams, 2020a). Researchers have been aware of these limitations for over half-a-century (Sundborg, 1956; Leopold et al., 1964), and have used empirically derived coefficients to account for the multiple scales of roughness present (Chow, 1959; Hey,





1979), although such an approach has considerable limitations (Ferguson, 2007; Adams, 2020a). There have also been more mechanistic attempts to disaggregate flow resistance, primarily into 'grain' (small-scale) and 'form' (large-scale) components (Parker and Peterson, 1980; Prestegaard, 1983; Hey, 1988; Weichert et al., 2009).

Advances in remote-sensing and statistics have allowed researchers to explore detailed scaling characteristics of gravel-bed surfaces using structure-functions (Furbish, 1987; Robert and Richards, 1988; Robert, 1991; Clifford et al., 1992), filtering (Bergeron, 1996), and transforms (Nyander et al., 2003). This approach to analyzing river beds has led to multiscalar decompositions of geometric roughness, rather than direct decompositions of hydraulic roughness. The latter approach has been developed for complex aeolian surfaces using transforms (Nield et al., 2013; Pelletier and Field, 2016; Field and Pelletier, 2018), which serves as a proof-of-concept for the multiscalar drag decomposition approach.

Also, as high-resolution spatial information becomes increasingly available to geomorphologists working in both laboratory and field environments (Westoby et al., 2012), there is a general need for statistical representations that effectively summarise these large datasets in a way that is informed by theory. Moreover, as energy-balance is increasingly recognized as a condition governing channel behaviour (Eaton and Church, 2004, 2009; Nanson and Huang, 2008, 2018; Church, 2015), improved understanding of flow resistance may contribute to a broader understanding of fluvial systems.

In a review of flow resistance in gravel-bed rivers, Adams (2020a) identified two relatively recent advancements in the fields of statistics and fluid dynamics that could contribute to a multiscalar drag decomposition tool. The first advancement is the wavelet transform, which is generally superior to the Fourier transform when analysing the underlying structure of complex and aperiodic signals, due to its use of a finite analysing function ('the wavelet') rather than a continuous one (Torrence and Compo, 1998). There are now various types of wavelet transform suited to different applications, some of which have been applied in rivers (Kumar and Foufoula-Georgiou, 1997; Nyander, 2004; Keylock et al., 2014). The second advancement is the development of roughness correlations for irregular surfaces (e.g. Forooghi et al., 2017), which estimate the fluid drag generated by a surface based purely on its geometric characteristics.

In this piece, we present a novel method of estimating the relative contribution of different physical scales of river bed topography to the total drag, using only a single longitudinal profile. The general approach consists of (1) a wavelet transform in which the channel surface is decomposed into a set of more simple components each at a different wavelength, and (2) a roughness correlation that estimates the drag associated with each wavelength, which is expressed as the equivalent sand roughness parameter $k_s$ (Nikuradse, 1933; Schlichting, 1936). By modifying the specific roughness correlation that is used, the transform-roughness correlation (TRC) approach may be applied in a wide range of hydraulic conditions. To demonstrate this tool, we present code in R language and apply it to a series of original laboratory experiments with high-resolution digital elevation models (DEMs), as well as some additional published data. Multiscalar drag decomposition provides researchers with useful information as they approach challenges pertaining to flow resistance and channel morphodynamics.



## 2 Methodological considerations

The transform-roughness correlation approach is a generic tool that should be adapted based on the hydraulic conditions and the purpose of its application. These considerations should span the data that is used, the type of wavelet transform, and the specific roughness correlation that is selected. We discuss these general considerations first as they provide context for the TRC approach.

First, the minimum resolution and spatial extent of the topographic dataset should be informed by the scale of the features of interest. The data should have a sufficiently small spatial resolution such that it can capture the range of bed features that contribute to drag. Also, to capture the characteristic geometry of bed features, the spatial extent of the dataset should be at least the length of the largest features that produce significant drag, for example, it should span many dunes crests or pool-riffle pairs.

Second, given that hydraulic roughness is of interest, the data could be reduced to streamlines representing primary flow paths. In some contexts, it may be acceptable to simplify the in-channel area to a one-dimensional profile extending along the thalweg, given that this should represent the surface that most of the flow interacts with. Alternatively, if the range of interactions between the flow and the surface is of interest, multiple streamlines (parallel or even intersecting) could be employed.

Third, the choice of wavelet transform in this context is a trade-off between the resolution of the decomposition and the ability to interpret it. The maximal overlap discrete wavelet transform (MODWT) offers several advantages over the discrete wavelet transform (DWT) and facilitates alignment between the original signal and the decomposition. The continuous wavelet transform (CWT) extracts more intricate structural characteristics from the original data and yields a greater number of wavelengths between which information is shared (Addison, 2018). However, the CWT generates a more abstract representation of the topographic variation at a given wavelength. A comparison of wavelengths extracted using MODWT and CWT methods is presented in Figure 1. At the wavelength corresponding to the spacing of a pool-bar-riffle sequence ($\lambda \approx 2$ m), the MODWT is aligned with the original thalweg elevation profile, but the CWT is not. Given that the CWT wavelengths do not resemble the original data, interpreting the results of roughness correlation applied to these wavelengths may be more difficult, and such results may be entirely invalid.

Fourth, the specific roughness correlation should match the channel's boundary Reynolds number $Re^* = U^*k/v$, where $U^*$ is shear velocity, $k$ is some representative roughness scale, and $v$ is kinematic viscosity. Given that the fluid dynamics characteristic of different boundary Reynolds numbers are highly varied, roughness correlations have only been developed for relatively limited ranges of $Re^*$. For example, given that gravel-bed rivers tend to be within the fully rough regime where $Re^* \geq 70$ (Buffington and Montgomery, 1997; Schlichting, 1979), it may only be valid to apply roughness correlations obtained for that regime specifically. Also, the flow should be turbulent, and it should be two-dimensional, indicated by flow aspect ratios ($w/h$, where $w$ is the wetted width and $h$ is flow depth) greater than 5 (Nezu and Nakagawa, 1993).

Last, roughness correlations in fluid dynamics tend to be developed for flows sufficiently deep to have logarithmic velocity profiles, which should be considered when they are applied to flows with less developed profiles. Jimenez (2004) suggested that logarithmic layers develop where relative submergence $h/k$ is greater than 40, although Cameron et al. (2017) observed





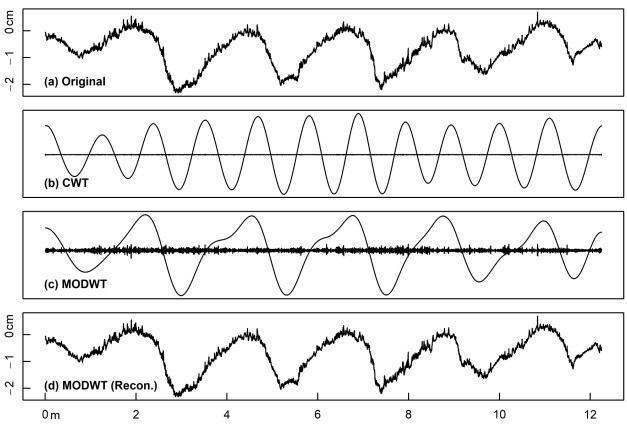

**Figure 1.** a) Thalweg elevation profile at end of Experiment 1a featuring a prominent pool-riffle sequence, where the x-axis represents distance upstream, b) grain ($\lambda$ = 4 mm) and form ($\lambda \approx$ 2 m) wavelengths derived from CWT, c) the same two wavelengths derived from a MODWT, and d) the original signal reconstructed from the MODWT by recombining wavelengths.

a logarithmic layer in rough open-channel flow at submergences as low as 1.9. During most flow conditions, it is common for gravel-bed rivers to have relative submergences of less than 10, and in some cases, as low as 0.1 (Lee and Ferguson, 2002; Ferguson, 2007), where no logarithmic layer can develop because roughness elements are not submerged. However, if one is interested in channel-forming flows capable of reworking the bed surface (Ashworth and Ferguson, 1989; Wolman and Miller, 1960) where relative submergence may be two orders of magnitude higher (Limerinos, 1970; Griffiths, 1981; Bray, 1982;
Millar, 1999), the logarithmic assumption should be satisfied for most rivers.

## 3    Application of TRC approach in gravel-bed rivers

### 3.1    Stream table experiment

To demonstrate the TRC approach, we required a large set of DEMs and associated hydraulic data for validation. We conducted a set of experiments using the Adjustable-Boundary Experimental System (A-BES) at the University of British Columbia
(Figure 2). The A-BES comprises a 1.75 m wide by 12.2 m long tilting stream table, and a recirculating water pump. The experiments were run as generic Froude-scaled models with an initial bed slope of 2 percent and a length scale ratio of 1:25, based on field measurements from steep gravel-bed rivers in Alberta, Canada. The bulk material ranged from 0.25 to 8 mm ($D_{max}$), with a $D_{50}$ of 1.6 mm and $D_{90}$ of 3.9 mm (see MacKenzie and Eaton, 2017). The banks were lined with roughly-cast interlocking concrete bricks to make a straight channel. Ten stream gages were equally spaced along the inner edge of the
bricks.





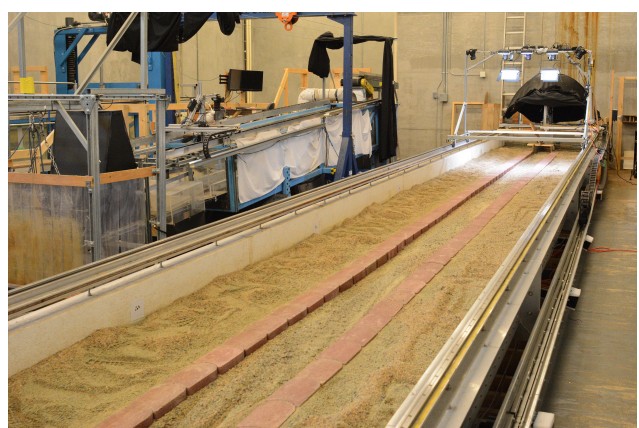

**Figure 2.** Adjustable-Boundary Experimental System (A-BES) at the University of British Columbia, showing the camera rig and the 30 cm wide channel configuration.

**Table 1.** Summary of experimental conditions in the A-BES.

| Run | Width $W$ (m) ($\pm$ 0.02) | Discharge $Q$ (L/s) ($\pm$ 0.03) | Duration (hrs) | DEMs |
|---|---|---|---|---|
| Exp1a | 0.3 | 1.5 | 16 | 24 |
| Exp1b | 0.3 | 1.0 | 16 | 24 |
| Exp1c | 0.3 | 0.67, 1.0, 1.5, 2.25 | 8, 4, 4, 4 | 68 |
| Exp2a | 0.08 | 0.4 | 16 | 24 |
| Exp2b | 0.08 | 0.27 | 16 | 24 |

### 3.1.1 Experimental procedure

The bricks were configured to make two different widths: (1) a 30 cm wide configuration that represents the scaled bankfull width of the field prototype, and (2) an 8 cm wide configuration which was selected based on preliminary experiments where channel width was decreased until bar formation was suppressed entirely. Thus, the two widths yield a range of bed morphologies and hydraulic conditions. A set of experiments were carried out for each configuration (Table 1). The first experiment ('a') consisted of a bankfull equivalent flow for the prototype for 16 hours, where the discharge was scaled with the width of the experimental channel $W$. The second experiment ('b') consisted of a flow two-thirds of the bankfull equivalent for 16 hours. The third experiment ('c'), conducted for the 30 cm wide channel only, consisted of low flow for 8 hours, and then three 4 hour phases with discharge increasing by a factor of 1.5 each time.

Before each experiment, the bulk material was hand-mixed to minimize downstream and lateral sorting, and the channel area was screeded to the height of the weirs at the upstream and downstream end. The flow was run at a low rate until the bed was fully saturated, and was then increased to the target flow. Each period of constant discharge was divided into phases of increasing duration, between which the bed was drained and photographed. Phases for the 16-hour experiments consisted of 5,



**Table 2.** Summary of A-BES experimental data collected during the final portion of each experimental phase. Values represent the mean of the last five measurements. $Re^*$ was calculated with $k = k^*_{s,pred}$.

| Exp | $W$ | $Q$ | $h$ | $F_r$ | $U$ | $U^*$ | $U/U^*$ | $\sigma_z$ | $h/\sigma_z$ | $Re^*$ | $k^*_{s,pred}$ | $h/k^*_{s,pred}$ |
|---|---|---|---|---|---|---|---|---|---|---|---|---|
| Exp1a | 0.30 | 1.50 | 0.014 | 0.95 | 0.36 | 0.053 | 6.76 | 0.0057 | 2.53 | 136 | 0.0034 | 4.32 |
| Exp1b | 0.30 | 1.00 | 0.012 | 0.85 | 0.29 | 0.049 | 6.00 | 0.0056 | 2.17 | 120 | 0.0032 | 3.72 |
| Exp1c(1) | 0.30 | 0.67 | 0.012 | 0.59 | 0.20 | 0.048 | 4.17 | 0.0051 | 2.27 | 105 | 0.0029 | 4.10 |
| Exp1c(2) | 0.30 | 1.00 | 0.013 | 0.70 | 0.25 | 0.051 | 4.96 | 0.0067 | 2.02 | 152 | 0.0039 | 3.47 |
| Exp1c(3) | 0.30 | 1.50 | 0.014 | 0.90 | 0.33 | 0.052 | 6.40 | 0.0063 | 2.24 | 154 | 0.0039 | 3.65 |
| Exp1c(4) | 0.30 | 2.25 | 0.018 | 1.03 | 0.43 | 0.060 | 7.28 | 0.0033 | 5.49 | 86 | 0.0019 | 9.79 |
| Exp2a | 0.08 | 0.40 | 0.015 | 0.92 | 0.35 | 0.054 | 6.47 | 0.0025 | 6.94 | 73 | 0.0018 | 11.55 |
| Exp2b | 0.08 | 0.27 | 0.013 | 0.75 | 0.26 | 0.050 | 5.29 | 0.0019 | 6.96 | 76 | 0.0020 | 7.34 |

10, 15, 30, 60, and 120 minutes, with four repeats of each. The 4 and 8 hour periods of constant discharge followed the same sequence but did not include the longest phases. In the final 30 seconds of each phase, the water surface elevation was recorded at each gage to the nearest 1 mm. Given that most of the maximum flow depths that were measured were greater than 15-20 mm, this degree of precision yields errors of approximately ± 10 percent.

The camera rig consisted of five Canon EOS Rebel T6i DSLRs with EF-S 18-55 mm lenses positioned at oblique angles in the cross-stream direction to maximise coverage of the bed, and five LED lights. Photos were taken in RAW format at 20 cm intervals, yielding a stereographic overlap of over two-thirds. Throughout the experiment, sediment collected in the trap was drained of excess water, weighed wet to the nearest 0.2 kg, placed on the conveyor belt at the upstream end, and recirculated at the same rate it was output. Zero sediment was fed into the system during the first 5-minute phase. For the five- and ten-minute phases, recirculation occurred at the end of the phase, and for the phases of longer duration, recirculation occurred every 15 minutes regardless of whether the bed was drained.

### 3.1.2 Data processing

Using the images, LAS point clouds were produced with Agisoft MetaShape Professional 1.6.2 at the highest resolution, yielding an average point spacing of less than 0.5 mm. Twelve spatially-referenced control points (and additional unreferenced ones) were distributed throughout the A-BES, which placed photogrammetric reconstructions within a local coordinates system and aided in the photo-alignment process. The point clouds were imported into RStudio where inverse distance weighting was used to produce DEMs at 1 mm horizontal resolution. Despite the use of control points, the DEM contained a slight arch effect whereby the middle of the model was bowed upwards. This effect was first quantified by applying a quadratic function along the length of the bricks, which represent a linear reference elevation. The arch was then removed by determining correction values along the length of the DEM using the residuals, which were then applied across the width of the model.

The channel thalweg for each DEM was determined by first locating pool centroids using the lowest ten percent of elevations at each cross-section, and then using Gaussian kernel regression to smooth vertices between the centroids. An example of




Earth **Surface**
Dynamics
Discussions

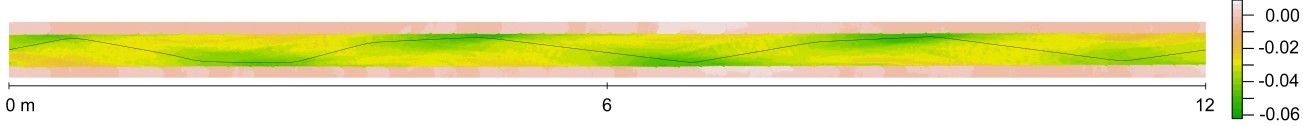

**Figure 3.** DEM of the channel morphology at the end of Experiment 1a, with estimated position of the thalweg. Zero represents the downstream extent of the model.

the estimated thalweg location is shown in Figure 3. For each DEM, ten wetted cross-sections were reconstructed using the water surface elevation data, which were used to estimate reach-averaged hydraulics. Mean hydraulic depth was calculated as $h = A/w$, where $A$ is cross-sectional area and $w$ is the wetted width. Velocity was estimated using the continuity equation $U = Q/A$. Shear velocity is $U^* = ghS^{1/2}$, where $g$ is gravity and $S$ is slope, and Froude number $Fr = U/(gh)^{1/2}$. A summary
of this data is presented in Table 2.

### 3.1.3 Additional experiments

In addition to the experiments conducted for this study, we obtained topographic and hydraulic data for 86 step-pool experiments published by Hohermuth and Weitbrecht (2018). The experiments were conducted in a 1:20 Froude-scaled model of a mountain stream, utilizing a range of bed slopes (8 – 11 percent), channel widths (15 – 35 cm), and discharges. For a given
experiment, a range of potentially usable elevation profiles were identified based on criteria for erroneous values, then the profile closest to the channel centreline was selected. Of the 86 experiments conducted, 83 experiments are used in this study. Thus, there is a total of 247 DEMs with associated hydraulic data when combined with the A-BES experiments.

### 3.2 The transform-roughness correlation approach

Here we specifically tailor the TRC approach to the geometric and hydraulic characteristics of gravel-bed channels. First, a
155 MODWT is applied to the thalweg elevation profiles of each DEM, yielding a set of simplified profiles representing topographic variation occurring at different wavelengths. Second, a roughness correlation for irregular surfaces in the fully rough regime is applied to each wavelength to estimate the associated drag. Using direct numerical simulation (DNS) in closed channels with systematically generated surface geometries, Forooghi et al. (2017) proposed the following empirical relation

$$\frac{k_s}{k} = F(Sk, \Delta) \cdot F(ES) \tag{1}$$

where $k$ is a measure of roughness peak heights, and $Sk$ is the skewness of the probability distribution of elevations. The functions $F(Sk, \Delta)$, $F(Sk)$, and $F(ES)$ are defined, respectively, as

$$F(Sk, \Delta) = \begin{cases} F(Sk), & \Delta \geq 0.35 \\ F(Sk)(1 + m(Sk) \cdot (\Delta - \Delta_0)), & \Delta \leq 0.35 \end{cases} \tag{2}$$





$$F(Sk) = 0.67Sk^2 + 0.93Sk + 1.3 \tag{3}$$

and

$$F(ES) = 1.05 \cdot (1 - e^{-3.8 \cdot ES}) \tag{4}$$

where $\Delta$ is a measure of the diversity of roughness peak heights ($\Delta = 0$ if peak heights are identical), $\Delta_0 = 0.35$, and $m(Sk) = 1.47Sk^2 - 1.35Sk - 0.66$. The parameter $ES$ is the effective slope which may be interpreted as the mean gradient of the local roughness elements (Napoli et al., 2008), and is given by

$$ES = \frac{1}{L} \int_L \left| \frac{dz(x)}{dx} \right| dx \tag{5}$$

where $z(x)$ is the height array, $x$ is the streamwise direction, and $L$ is the surface length in $x$. Effective slope is approximately proportional to drag in the range $0 < ES < 0.35$ (Napoli et al., 2008; Schultz and Flack, 2009). In the TRC approach, calculating the $\Delta$ parameter is impractical given that longer topographic wavelengths may contain very few (or even one) complete oscillations that could be interpreted as roughness peaks. As a result, we simply use the $F(Sk)$ term in Equation 2, which is likely appropriate given that for most natural surfaces $\Delta \gg 0$ (Forooghi et al., 2017). In the original equation, $k$ was defined as the peak-to-trough height of the surfaces, however, we adopt the standard deviation of elevations $\sigma_z$ for the natural surfaces analysed herein as they are more topographically variable than the numerically-generated surfaces, and there is less bias towards extreme peaks and troughs. The effects of different choices of $k$ are briefly explored in Figure 7. The estimated drag for each wavelength is expressed as $k_s$. The experimental data and code that performs the MODWT and applies the roughness correlation is available online. In the following section, we present the results of the TRC approach applied to the experiments.

## 4 Results and Discussion

In this section, we primarily focus on the results from Experiment 1a, which features a well-developed pool-riffle-bar sequence formed at a bankfull flow. First, we show how the key parameters of the roughness correlation (effective slope and standard deviation) vary across each wavelength. Second, we estimate the relative contribution of different scales of bed topography to the total drag and explain how the estimated values relate to the key parameters and the characteristics of the experimental surfaces. Third, we perform a sensitivity analysis for the choice of analysing function (known as the 'mother wavelet') in the MODWT and the choice of $k$ in the roughness correlation. Finally, using all of the available experiments, we demonstrate that the roughness correlation may be used to estimate total flow resistance using a conventional equation.

Earth **Surface**
**Dynamics**
Discussions

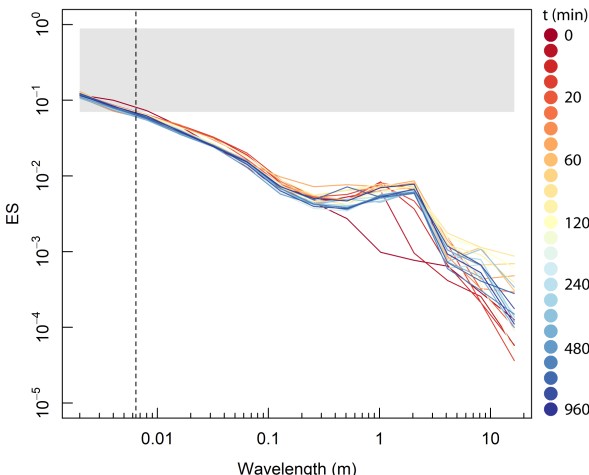

**Figure 4.** Effective slope of each topographic wavelength during Experiment 1a, where each line represents a point in time. Vertical dashed line represents the largest grain diameter in the experiment. The shaded area represents the range of $ES$ values of the surfaces generated by Forooghi et al. (2017).

Figure 4 shows the effective slope of each topographic wavelength (or spatial scale) over the course of Experiment 1a. The effective slope is greatest at the grain scale wavelengths ($\lambda \leq D_{max}$) where the surface is locally rough and reduces as the wavelength increases and the topography is more locally subdued. This is demonstrated in Figure 1b, where wavelengths at the grain scale have more acute oscillations than those at longer wavelengths. The main exception to this trend is the wavelength of around 2 m where there is a prominent peak in the ES distribution, associated with the development of the pool-riffle-bar

sequence approximately ten minutes into the experiment. It is important to note that most of the topographic wavelengths have values of $ES$ (and $k_s/k$ in Equation 1) that are smaller than the surfaces used by Forooghi et al. (2017) to develop the roughness correlation. Thus, it may be more appropriate to focus on the relative values of $k_{s,pred}$ for a specific combination of channel geometry and roughness correlation.

Figure 5a shows the standard deviation of each topographic wavelength for Experiment 1a. Except for the first ten minutes

(i.e. first two DEMs) during which the bed morphology is developing, there is a minor peak in $\sigma_z$ at the scale of 3 cm, and a major peak at the scale of 2 m. At the smallest scales, topographic variation tends towards zero, and there is some contribution to $\sigma_z$ at the largest scale due to the slightly concave or convex shape of the profile. Figure 5b presents this data as a cumulative percentage, which shows that the grain scale accounts for less than five percent of all topographic variation. This figure is the same as the Form Size Distribution (FSD) proposed by Nyander et al. (2003).

Using the TRC approach, we present the distribution of $k_s$ predicted using Equation 1 in Figure 6a. Following the format of 'grain size distribution' and 'form size distribution', we term this style of plot the drag size distribution (DSD). There is




Earth **Surface**
**Dynamics**
Discussions

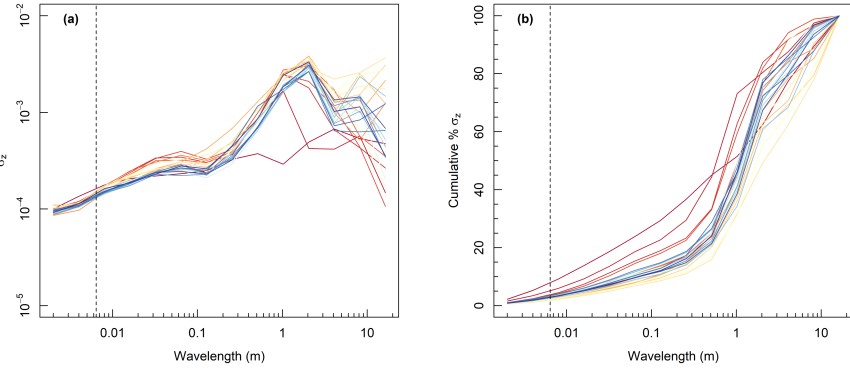

**Figure 5.** Form size distribution during Experiment 1a. Standard deviation of each topographic wavelength presented as an (a) absolute, and (b) cumulative percentage, for longitudinal profiles during the experiment. Refer to Figure 4 for legend.

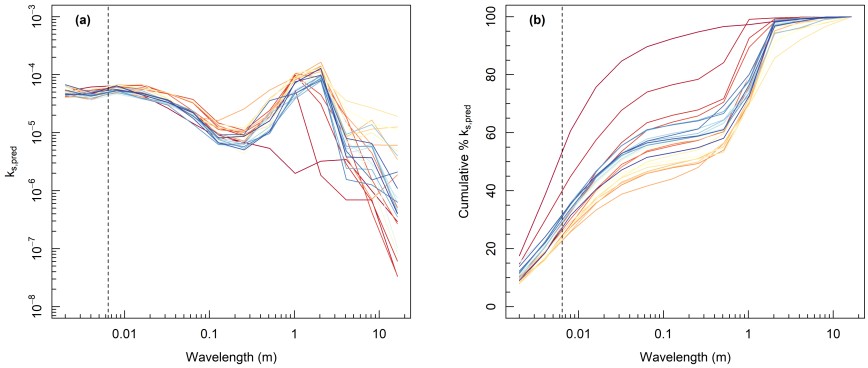

**Figure 6.** Drag size distribution during Experiment 1a. Estimated drag associated with each topographic wavelength presented as an (a) absolute, and (b) cumulative percentage, for longitudinal profiles during the experiment. Refer to Figure 4 for legend. Note that the absolute values of $k_{s,pred}$ appear unusually small for the surfaces ($\ll$ 1 mm) compared to values of $\sigma_z$, which is discussed later.

a major peak in the DSD at the scale of 2 m, and a minor peak at the scale of approximately 5 mm (around the size of the largest grains). At small scales, and large scales especially, estimated $k_s$ tends downwards. Figure 6b presents the DSD as a cumulative percentage, which shows that the $k_s$ associated with the grain scale is estimated to account for approximately
30 percent of the total $k_s$. This proportion of grain- and form-drag is similar to estimates in gravel-bed rivers with similar morphologies (Hey, 1988; Parker and Peterson, 1980; Prestegaard, 1983), which indicates that the TRC approach provides physically realistic estimates of drag.





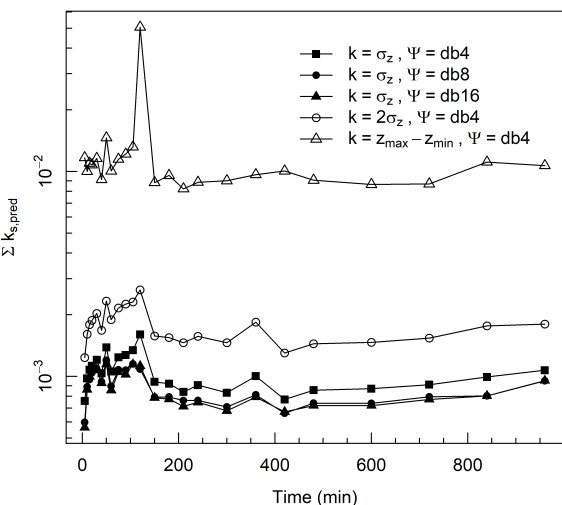

**Figure 7.** Sensitivity analysis of $\Sigma k_{s,pred}$ to choice of mother wavelet $\Psi$ in MODWT and $k$ in the roughness correlation for Experiment 1a, where $z_{max} - z_{min}$ is the maximum range of elevations. The combination of $\sigma_z$ and Daubechies 4 wavelet is used in this study.

Figure 7 demonstrates the dependency of $\Sigma k_{s,pred}$ (i.e. sum of $k_{s,pred}$ for a given DEM) on the choice of mother wavelet and the $k$ value in the roughness correlation. Various mother wavelets from the Daubechies family have been used when applying
wavelet transform to river bed topography (Nyander et al., 2003; Gutierrez et al., 2013; Qin et al., 2015), and for the same $k$ value, these mother wavelets yield similar results. The values of $\Sigma k_{s,pred}$ are more sensitive to the choice of $k$. If standard deviation is used as the estimate of $k$, whilst changing $\Psi$, there is a relatively similar pattern of $\Sigma k_{s,pred}$. If the absolute range of elevations is used as $k$, as was used by (Forooghi et al., 2017), extreme outliers in the elevation profile disrupt a physically realistic pattern of total drag across the experiments, which should consist of at least an initial increase as the pool-bar-riffle
sequence emerges.

The approach of adding up the effect of different roughness elements to estimate a net effect, although established in the literature (Cowan, 1956; Einstein and Banks, 1950; Hey, 1988; Leopold et al., 1960; Millar, 1999), has been demonstrated to have limitations given that superimposed combinations of different roughness elements may produce drag feedbacks in either direction (Li, 2009; Wilcox and Wohl, 2006). Thus, it is important to demonstrate that $\Sigma k_{s,pred}$ is proportional to the total $k_s$.
Using data from all experiments conducted for this study, in addition to the step-pool experiments of Hohermuth and Weitbrecht (2018), Figure 8 compares the relationship between $\Sigma k_{s,pred}$ and the $k^*_{s,pred}$, the latter being the estimate of $k_s$ obtained by applying the roughness correlation to the elevation profile without the transform. Each of the two datasets are described by a linear relationship, which demonstrates that both the transform and non-transform estimates of $k_s$ are proportional. This validates the TRC approach for a given dataset, however, it is worth noting that the two datasets are characterised by different



Earth **Surface**
**Dynamics**
Discussions

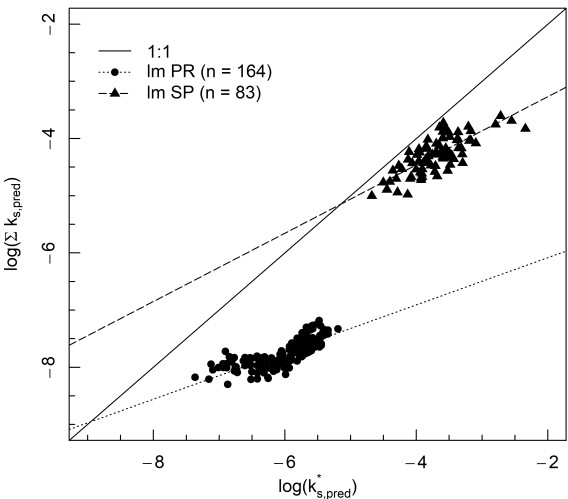

**Figure 8.** Relationship between $k^*_{s,pred}$ and $\Sigma k_{s,pred}$ for the pool-riffle (PR) experiments carried out for this study, as well as the step-pool (SP) experiments conducted by Hohermuth and Weitbrecht (2018).

slopes. The difference in slope likely arises due to the specific characteristics of each topographic dataset, which affect the wavelet decomposition of the wavelengths, and in turn, the values of $k_{s,pred}$. The choice of $k$ in the roughness correlation does not affect the linear relationship between $k^*_{s,pred}$ and $\Sigma k_{s,pred}$.

Figure 9 compares the relationship between estimated flow resistance $(8/f)^{1/2}$ and relative submergence $h/k$, using two different values of $k$. In Figure 9a, relative submergence is calculated using $\sigma_z$, which is now common in gravel-bed rivers,
whereas in Figure 9b, it is calculated using $k^*_{s,pred}$. Fergusons's (2007) variable-power equation (Appendix A) is applied with coefficients reported by Chen et al. (2020), which were fitted to a wide range of gravel-bed channels using $k = \sigma_z$. It is interesting to note that the $k^*_{s,pred}$ approach to relative submergence yields a closer fit to the Chen relation, quantified by a 30 percent smaller root-mean-square error. This result suggests that the estimates of $k_s$ based on Equation 1 are useful in predicting flow resistance in rivers, and thus provides evidence for the multiscalar drag decomposition.

**5   Implications for flow resistance in rivers**

The roughness correlation developed by Forooghi et al. (2017) incorporates information regarding both the height of the roughness elements (a vertical roughness scale, Nikora et al. 1998) and the arrangement or spacing of roughness elements (a horizontal roughness scale, Bertin and Friedrich 2014). In isolation, either of these roughness scales may be misleading. For example, effective slope is a horizontal roughness metric and can be proportional to drag for some surfaces (Napoli et al.,



Earth **Surface**
**Dynamics**
Discussions

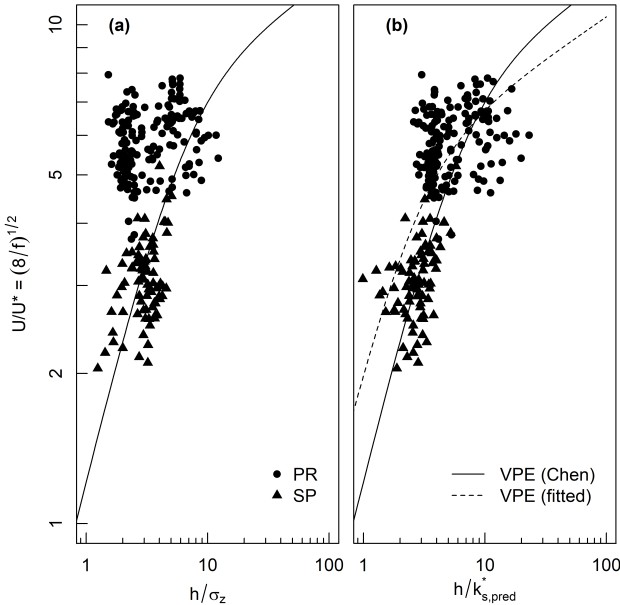

**Figure 9.** Plot of $(8/f)^{1/2}$ against relative submergence $h/k$, where (a) $k = \sigma_z$, and (b) $k = k_{s,pred}^*$. Solid line is the Ferguson (2007) variable-power equation (VPE) using coefficients $a_1 = 5.77$ and $a_2 = 1.24$ (Chen et al., 2020). Root-mean-square errors are 2.03 and 1.42, respectively. Dashed line is the VPE fitted to the data, where $a_1 = 4.81$ and $a_2 = 2.36$ (note: VPE could not be fitted to $h/\sigma_z$).

2008; Schultz and Flack, 2009). Thus, in isolation, Figure 4 would indicate that the small-scale bed features are most effective at producing drag. Alternatively, the standard deviation of surface elevations is a vertical roughness metric and has also shown to be proportional to flow resistance (Aberle et al., 1999; Chen et al., 2020). Therefore, in isolation, Figure 5 could lead to the interpretation that only the largest scale bed features produce drag. It is important to recognize that depending on the surface of interest, drag is usually a compromise between vertical and horizontal roughness, which is inherent in Equation 1 and the

drag size distribution (Figure 6).

In gravel-bed rivers, which are typically ungauged, and where measurement of hydraulic variables is subject to practical limitations (Miller, 1958), flow resistance is typically estimated using only a vertical roughness scale. Upon introducing his method of sampling coarse bed material, Wolman (1954) remarked that these data could be used to estimate hydraulic rough-ness, and Lane (1957) agreed, based on the notion that grain diameter represents a vertical roughness scale as demonstrated

by Nikuradse (1933). However, the relationship between grain diameter and flow resistance breaks down in natural channels for two main reasons: (1) grain diameter does not account for larger and often more hydrodynamically significant roughness elements (Sundborg, 1956; Leopold et al., 1964; Bathurst, 1982), and (2) it does not consider the horizontal spacing of these larger roughness elements (Schlichting, 1936; Nowell and Church, 1979; Davies and Sutherland, 1980), which has a systematic effect on drag (Morris, 1955; Leonardi et al., 2007; Napoli et al., 2008).



In recent years, the increased availability of high-resolution topographic data has led to the adoption of $\sigma_z$ as a roughness scale, on the basis that it accounts for larger-scale bed structures (Aberle et al., 1999; Aberle and Smart, 2003; Cadol and Wohl, 2013; Smart et al., 2002; Yochum et al., 2014). However, $\sigma_z$ only improves upon the first deficiency of grain-based roughness metrics, and consequently, it cannot be considered a measure of hydraulic roughness. The roughness correlation used herein may be a significant improvement over existing roughness metrics as it incorporates both vertical and horizontal roughness
scales, and provides a direct semi-empirical estimate of $k_s$. Moreover, roughness correlations of this variety can be applied to most datasets where $\sigma_z$ is calculated.

## 6    Conclusions

The transform-roughness correlation approach allows researchers to estimate the relative contribution of various scales of bed topography to the total drag. By modifying the roughness correlation to suit the hydraulic conditions, multiscalar drag de-
composition may be achieved in virtually any type of river, and perhaps boundary-layers in other environments. The only requirement is that the topographic data is of a sufficient resolution and spatial extent to capture the scales over which the hydraulically-significant roughness elements occur, and data of this quality is only becoming more available to geomorphologists over time. In particular, we expect that given the continual advances in methods for collecting bathymetric data in both shallow (Kasvi et al., 2019) and deep channels (Dietrich, 2017), applying the TRC approach will become increasingly practical
in natural rivers.

Given that the TRC approach may provide new and more detailed information regarding the effect of bed geometry on fluid dynamics, incorporating both horizontal and vertical scales of roughness, it may contribute to advances in hydraulics as well as an understanding of channel morphodynamics. The estimates of total $k_s$ from semi-empirical roughness correlations may present more immediate benefits by serving as replacements for representative roughness values, which have historically been
necessitated by technological limitations.

The application of the TRC approach herein demonstrates the limitations of commonly-used approaches to estimating flow resistance in rivers, which rely solely on representations of vertical roughness and ignore their horizontal arrangement. Also, it highlights the utility of wavelet transform as a tool that provides intuitive representations of channel bed topography. The TRC approach is currently being used to explore channel morphodynamics and bedload transport using laboratory experiments.

*Code and data availability.*   Data and code are available online (https://doi.org/10.5281/zenodo.3879652; Adams (2020b)).



Earth **Surface**
Dynamics
Discussions



## Appendix A: Ferguson (2007) variable-power equation

Ferguson (2007) presented the variable-power flow resistance equation

$$(8/f)^{1/2} = \frac{a_1 a_2 (h/k)}{(a_1^2 + a_2^2 (h/k)^{5/3})^{1/2}} \tag{A1}$$

where $a_1$ and $a_2$ are empirically-derived coefficients, $h$ is flow depth or hydraulic radius, and $k$ is some representative
roughness scale.

*Author contributions.* DLA is responsible for conceptualization, investigation, formal analysis, and writing. AZ provided expertise in open-channel flow, contributing to the interpretation and communication of the results and the proposed technique.

*Competing interests.* The authors declare that they have no conflict of interest.

*Acknowledgements.* The authors would like to thank William Booker, Lucy MacKenzie, Brett Eaton, and Ian Rutherfurd for reviewing
the original manuscript, and Benjamin Hohermuth for providing the laboratory step-pool data. This work was supported by postgraduate
scholarships provided to DLA by the Australian and Canadian Governments.



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
