# Peer review of "Short communication: Multiscalar roughness-length decomposition in fluvial systems using a transform-roughness correlation (TRC) approach"

_Earth Surface Dynamics, 2020_

## Referee Comment (RC1) · Anonymous Referee #1 · 30 Jul 2020

The manuscript entitled "Multiscalar drag decomposition in fluvial systems using a transform-roughness correlation (TRC) approach", submitted for publication as a short communication to Earth Surface Dynamics presents an interesting study regarding the determination of flow resistance in fluvial channels. The presented method is based on a the thalweg profile and good results are presented by the authors. Although the topic of the manuscript fits the scope of the journal, I found that it is not suitable for publication in its present form due to various reasons. For example, the authors need to expand their consideration towards the heterogeneity of the surface (and not only a

single profile). In fact, focusing solely on the thalweg profile means that some morphological features such as banks etc. are not adequately captured through the analysis. This is, however, only one major issue that I identified, as becomes apparent from my comments below, in which I also comment on the experimental data, experimental procedure, theoretical background, data analysis, data interpretation, and the number of references. Overall, the approach is certainly interesting, but it cannot be presented in the form of a short communication (in my opinion), as there are simply too many open questions and too many shortcuts in the manuscript. I therefore recommend rejection of the manuscript and resubmission as a research paper.

Detailed comments: L11: Having read the communication, I am not sure what conventional equation is meant by the authors. L22: It is stated that the grains and bedforms on the surface span orders of magnitude of scales. So the question arises if the thalweg profile is really sufficient to capture all spatial scales? (see also L48) L50: The reference for the used roughness correlation should be given here, as it was not really developed by the authors. L62: There are many morphological features that contribute to drag which are not considered when analyzing the thalweg profile (this includes the curvature of the channel, alternate bars and many more). This needs to be highlighted better. L64: It is stated that the spatial extent of the data should at least cover the largest features that produce significant drag (resistance). A dune is one such feature and it becomes not really clear why many dune crests need to be included in the dataset. This should be elaborated in some more depth. In this context, can the spatial extent of 3D-features really be described using a single profile? L66: The data meant by the authors should be specified, as I am not sure how data can be reduced to streamlines. A streamline is a line that follows the direction of flow velocity and is a hydraulic feature. So how exactly can topographic data be reduced to a streamline? Also, how can streamlines intersect when these lines follow the direction of flow? This should be specified. L70: I fully acknowledge that the authors use wavelet approach. However, the principle of wavelets should be described in some more depth, as not all reader will be familiar with the principles of the wavelet transforms described here.

L76: Fig. 1 presents data from Experiment 1a - these data have not been described and no reference is found to an experimental study. This is confusing. For example, how does the reader know the shown profile originates from a riffle-bar sequence and that the MODWT is aligned with the thalweg elevation profile? In this context, why is the CWT not aligned with the thalweg-profile? It is also a wavelet transform that is based on the thalweg-profile - so it should be aligned with this profile? This could be formulated more clearly. L84: I fully agree but would also argue that almost all gravel bed rivers are in the hydraulic rough regime. In this context, what is meant by the statement that roughness correlations have only been developed for limited ranges of Re*? Typically, three ranges are distinguished, and existing approaches can be found that cover all these ranges. L85: I am not sure that the width to depth ratio is an appropriate measure to determine if the flow is 2D. For example, the flow in the roughness layer of gravel beds is far from being 2D. Note also 2D conditions also depend on the relative submergence (I acknowledge that this is mentioned in the next paragraph, but I would expect such a statement already here). L95: In this case I could also agree with the 2D-statement from the last paragraph. L96: The TRC approach remains a black box until here. L100: How was discharge measured? L101: How was sediment transport scaled? L104: What was the working principle of these gages and what was the distance between them? Investigating Table 2, I found that the water depths were very low (< 0.02 m). This means that surface tension can impose a significant scale-effect biasing the results. This needs to be discussed. In this context, what was the accuracy of the water depth readings? I am asking because the differences in water depth are within the millimeter range. L111: How exactly can discharge be scaled with the width of the experimental channel when using a Froude-scale model? The information that bankfull discharge was used is enough, but it should be stated how deep the channel was. Also, is there information available how much discharge was conveyed through the sediment bed (this could be important given the low discharges)? L116: Which weirs? Was there a backwater effect? L117: What was the corresponding flow rate? That would help to answer my question @L111? L118: Had the draining of the

bed any effect on the topography and was the bed saturated again when increasing the discharge after the measurements? Table 2 lacks of units and k*_s,pred has not been defined properly (see below). L121: Referring back to my comment @L104 - it is stated 1hat the water surface elevation was determined to the nearest millimeter and that the mean water depths are lower than the maximum water depth (what is meant by the statement "most of the maximum flow depths"?), the degree of precision is different than stated @L122. L141: Figure 3 shows the thalweg for the total length of the table of Exp1a and the corresponding profile is shown in Fig.1a. This means to me that there is an effect of the inflow and outflow section on the bed morphology (which is directly visible in Fig. 1a) which in turn may affect the results of the wavelet analysis. This needs to be evaluated. There is also a mismatch between the thalweg elevations the figures - please explain. Moreover, the thalweg is also meandering - how was this accounted for? Note also that it was mentioned that the bricks represent a linear reference elevation. Looking at the colors of the bricks and the color-scale of Fig. 3 I would actually disagree with this statement (note also that no units are given for the scale). Another comment concerns the wetted width which, according to the numbers presented in Table 2, was different, even for bankfull conditions. I calculated the wetted width for Exp1c(3) from the numbers presented in Table2 and the formulas given at L143 and 144 and obtained a wetted width of 0.32 m which is larger than the channel width of 0.30 m. How is that possible? This in turn raises some serious questions regarding the accuracy of the experimental data which needs to be discussed in much more depth given the small scale of the experiments (see also my comment regarding the surface tension). This, together with the influence of the inlet and outlet sections indicates that more work is required to substantiate the results of the study. L149: What was the range of the discharges and what was the grain-size distribution of the bed material? L157: Strictly speaking, drag is not estimated. What is estimated is k_s, a roughness length. L159: Since skewness plays an important role, it would be good to show corresponding distributions. L166: What exactly is meant by "diversity of roughness peak heights"? This remains unclear. L174: I see five peaks in the profile in

Fig.1 - this number is not sufficient? (I acknowledge my comment above regarding the influence of the inflow and outflow section). Nonetheless, this needs to be discussed in more detail. L175: What was the range of delta in the present investigation? L178: I don't understand Figure 7 from the previous presentation of the material. L179: But k_s is a length scale which is different from drag. Please clarify. L183 and following: See my comment regarding the influence of the in- and outlet sections on the morphology of the bed. L190: I assume that the profile was detrended for the analysis, i.e. the bed slope is not considered in this analysis? L197: What is meant by k_s,pred? How was this determined? L201: It is stated that topographic variation tends towards zero, but above (L191) it is stated that the effective slope is greatest at grain-scale wave lengths (this could also be seen as a measure of topographic variation at another scale). This is contradictory in my opinion. Please explain. L024: This figure is the same as the Form Size Distribution proposed by Nyander et al (2003)? How is it possible that it is the same? Please be more specific. L205: I am confused now. Here it is stated that Equation 1 is used to predict k_s, which is ok. But what is then the relative value of k_s,pred (see L 197)? I also see the need to define the concept of determining k_s for different wavelengths from a hydraulic point of view in more detail taking the physics into account (and the assumptions on which the determination of k_s is based: For example, were local values of velocities and slopes used? How was it ensured that uniform flow conditions prevailed? What about 2D-flow conditions at the grain scale? Don't get me wrong, the presented results are certainly interesting, but this needs to be elaborated in much more depth in my opinion. L213: The DEM is not analyzed, but the thalweg-profile extracted from the DEM. L214: The mother wavelet is only mentioned in the figure caption indicating the need to present the chosen approach in much more detail. This is rather confusing (also the choice of the other wavelets indicated in Figure 7 which remains a black box to me). L218: I would argue that I only see one extreme outlier. L224: I basically agree, and this would be one step towards answering my comment @L205. However, I am not sure that I understand the statement regarding the proportionality - it should give the same value. L226: I still do not understand exactly how k*_s,pred was obtained. What is meant without the transform? Does that mean the overall profile was used? This needs to be described in much more depth. L229: Why does that validate the TRC approach? The numbers deviate (see my comment @L224). A comment in between - all this compares (to my understanding) the presents results in regard to the approach of Forooghi et al. (2017) - but how does that approach relate to the real k_s value? That means what is the "real" k_s value from the experiments? This needs to be discussed in depth. It seems that the hydraulic data have not been used to determine k_s (I might be wrong here, but this indicates that a more precise presentation of the material is required). Figure 9: First, see my comments regarding the experimental data. Second, why is k*_s,pred used here and not sum(k_s,pred). This is confusing, as the latter parameter has been derived but is not presented in this final plot. L240 and following: Please consider my above comments. L251 and following: This main information here should have been presented in the introduction in my opinion. L268 and following: Please consider my above comments. References: The short communication is overloaded with references.

---

## Referee Comment (RC2) · Anonymous Referee #2 · 17 Aug 2020

General comments:

The manuscript describes and discusses a potentially useful new technique to assess flow resistance in natural channels and particularly in gravel-bed streams. The novel method proposes to estimate the relative contribution of different physical scales of river bed topography to the total drag. To do so, a transform-roughness correlation (TRC) approach is used, based on a wavelet transform and a correlation of different elements of the bed structure with a roughness metric. The approach is applied to two different sets of flume experiments, one reproducing a pool-riffle morphology and the

other a step-pool morphology.

General considerations that lead to the proposition of this new approach were reported in a recent paper of the first author (Adams, 2020a). The manuscript appears to be an attempt to apply this approach to "natural" gravel-bed channels as reproduced in flume experiments. Although the main results appear to provide some support for further exploring the suitability of this new approach, some ideas in the manuscript are not explained clearly enough or in sufficient detail, and some elements would warrant additional discussion in my view.

1. In the present study, only one single longitudinal profile is analysed. Although it is mentioned that "multiple streamlines (parallel or even intersecting) could be employed", I think that this aspect should be discussed in more detail. For example, how much would the results be affected by selecting different streamlines? How representative is a single streamline for the flow conditions averaged over the cross-section?

2. Frequent reference is made to the study of Forooghi et al. (2017), but for readers unfamiliar with this study it is not always clear what is meant exactly or what meaning for example the term "effective slope" had in this cited study (see also my specific comment below to L160-178). Therefore I suggest to provide some more information on this important background study.

3. The application of the TRC approach to the two different sets of flume experiments as illustrated in Figure 9 appears to result in a somewhat better flow resistance prediction than more traditional approaches (reduced root-mean-square error when using the $k^*_{s,pred}$ roughness measure as compared to using the $\sigma_z$ measure), if the measure $k^*_{s,pred}$ really refers to the application of the new TRC approach. However, it is not clear how $k^*_{s,pred}$ was calculated, this needs to be clarified.

4. It is known from other studies on flow resistance in gravel-bed streams that the presence of large wood on the streambed can substantially alter the total resistance, and that particularly in such a situation using $\sigma_z$ as compared to using e.g. a

characteristic grain size such as D84 improves flow resistance calculations. Can you speculate if and how using the TRC approach could further improve flow resistance calculations in such settings?

5. I see one potentially interesting further application of the TRC approach with regard to the question of stress partitioning in gravel-bed streams which is one important approach to improve bedload transport predictions in these channels (e.g. Ancey, 2020, JHR, part 2, https://doi.org/10.1080/00221686.2019.1702595). Some discussion of this aspect would be welcome.

Specific comments:

Figure 1b): indicate which line refers to grain and form wavelength

L160-178: In the process of selecting an appropriate correlation between roughness measures and elements of the wavelet analysis, the authors refer to the study of Forooghi et al. (2017) who used a variable called "delta" (a measure of the diversity of roughness peak heights), and report that "effective slope is approximately proportional to drag in the range 0 < ES < 0.35". It is not clear whether the authors also determined "delta" or not. Furthermore, in eq. (2) a critical value of delta = 0.35 is used to separate the two ranges, whereas later in the text a (critical) value of 0.35 is associated with ES (L172). This is all somewhat confusing and requires clarification.

Figure 4: If the vertical dashed line is meant to indicate Dmax, it should plot at 0.008 m (L102).

---

## Author Comment (AC1) · 27 Aug 2020

**General comments**

We would like to thank the reviewer for highlighting three aspects of the manuscript that required major modifications or additions. First, there were several areas where the presentation and interpretation of the TRC analysis was unclear, which stimulated us to modify the expression and provide a more clear and comprehensive explanation. Second, the description of the experimental methods provided was insufficient and detracted from the findings, which has compelled us to provide additional detail and importantly, a quantification of error. Third, upon recommendation, we calculated ks values from the hydraulic data using a Colebrook-White type equation, which allowed us to demonstrate that the roughness correlation provides fairly accurate estimates of the total ks (within a factor-of-two). First, general comments are presented and then followed by specific comments we feel appropriate to address at this time.

*Reviewer: [There are] some serious questions regarding the accuracy of the experimental data which needs to be discussed in much more depth given the small scale of the experiments (see also my comment regarding the surface tension). This, together with the influence of the inlet and outlet sections indicates that more work is required to substantiate the results of the study.*

Author: We would like to briefly respond to this comment, before a more in-depth response below. In the absence of error estimates provided in the initial submission, the reviewer has raised a reasonable doubt as to whether the dataset is of sufficient quality. From a practical standpoint, the small-scale of the model is necessary to fully replicate in-channel morphology in gravel-bed rivers, notably, bars, pools, and riffles. The superposition of both grain (small) and form-scale (large) resistance elements provides an optimal dataset for the demonstration of the TRC approach. However, the trade-off is an inevitable decrease in the degree of precision of the model, notably, the measurement of hydraulic quantities (due to shallow water depths and surface tension). As we demonstrate with a brief analysis of measurement error for the stream gauges, this does not compromise the results of the study. Based on the degree of measurement precision (1 mm), there is on average a plus-minus 8 percent error in estimates of flow depths. Also, the importance of edge-effects within the experiments have been over-stated, and we demonstrate with an analysis that they have little-to-no effect on the wavelet transform, nor do they affect the roughness length analysis.

*Reviewer: For example, the authors need to expand their consideration towards the heterogeneity of the surface (and not only a single profile). In fact, focusing solely on the thalweg profile means that some morphological features such as banks, etc. are not adequately captured through the analysis.*

Author: We agree, complete three-dimensional decomposition of the roughness length in the channel would include information regarding the banks, overall channel gradient, vegetation, etc., however, this is beyond the scope of this paper. The proposed approach pertains to only in-channel features and provides a more detailed and theoretically robust analysis compared to other existing approaches in rivers (notably, grain size, standard deviation of elevations). We will clarify that the TRC approach, in its present form, is an analysis of fluid drag associated with in-channel features and that it does not consider other resistance elements, nor three-dimensional interactions between flow and channel topography. Also, as demonstrated in the specific comments, the good correlation between topographic (using the roughness correlation) and hydraulic (using the Colebrook-White equation) validates the one-dimensional approach for the experiments used herein.

*Reviewer: I also see the need to define the concept of determining k_s for different wavelengths from a hydraulic point of view in more detail taking the physics into account*

Author: As discussed (L221-224), the disaggregation of flow resistance into different scales (flow superposition approach) has been a common conceptual and analytical approach since the 1950s, and the suggested approach in this study is merely an extension of this. Notable examples include (1) attempts to estimate the relative importance of grain and bedform resistance using statistics (L25-27, e.g. Weichert et al. 2009), and (2) the reverse, in which one may roughly estimate the total frictional resistance of a natural channel by noting the presence of specific resistance elements (e.g. Cowan 1956, Manning coefficients for natural channels). As discussed, Li (2009) and Wilcox and Wohl (2006) have emphasised that the combination of different roughness elements may lead to nonlinear drag feedbacks in either direction, however, the decomposition approach holds that roughness elements (e.g. a grain or a riffle) have relatively discrete and linearly additive hydraulic effects.

We are unable to provide an explanation beyond which has already provided in the literature; however, we have been able to demonstrate that the assumptions of the superposition approach hold, in two different ways. The first is comparing the estimates of ks by applying roughness correlation with and without the wavelet transform, and the second is by comparing these estimates of ks to the 'true' value of ks, calculated using the Colebrook-White approach. We expand on these two points below.

**Specific comments**

*Reviewer: L11: Having read the communication, I am not sure what conventional equation is meant by the authors.*

Author: We were referring to a conventional flow resistance equation using relative submergence, which will be clarified.

*Reviewer: L22: It is stated that the grains and bedforms on the surface span orders of magnitude of scales. So the question arises if the thalweg profile is really sufficient to capture all spatial scales? (see also L48)*

Author: If in-channel features are of interest, a profile may capture the one-dimensional character of the roughness elements if it is sufficiently long. For example, in the dataset presented herein, the thalweg profile spans 5-6 pool-riffle pairs (the dominant resistance elements), which is sufficient to capture their characteristic geometry (height, spacing). As explained in L61-65, the required length of the profile would be dependent on the features of interest.

*Reviewer: L50: The reference for the used roughness correlation should be given here, as it was not really developed by the authors.*

Author: We have provided the reference at this point.

*Reviewer: L62: There are many morphological features that contribute to drag which are not considered when analyzing the thalweg profile (this includes the curvature of the channel, alternate bars and many more). This needs to be highlighted better.*

Author: We agree, it is important to acknowledge that the analysis pertains to in-channel features only, as well as noting the potential importance of other roughness elements. See general comments.

*Reviewer: L64: It is stated that the spatial extent of the data should at least cover the largest features that produce significant drag (resistance). A dune is one such feature and it becomes not really clear why many dune crests need to be included in the dataset. This should be elaborated in some more depth. In this context, can the spatial extent of 3D-features really be described using a single profile?*

Author: From the sampling perspective, it is ideal to have a spatial extent that spans multiple roughness elements of the same type (riffles, dunes, steps) to establish some characteristic geometry (notably, the height of roughness elements and their spacing or slope). This has been clarified in the text. As discussed, three-dimensional effects are outside the scope of this paper.

*Reviewer: L66: The data meant by the authors should be specified, as I am not sure how data can be reduced to streamlines. A streamline is a line that follows the direction of flow velocity and is a hydraulic feature. So how exactly can topographic data be reduced to a streamline? Also, how can streamlines intersect when these lines follow the direction of flow? This should be specified.*

Author: We agree. We will remove mention of intersecting streamlines and will replace the word 'streamline' with 'thalweg elevation profile'. We have clarified that the thalweg elevation profile is used due to its association with the dominant vector of flow velocity.

*Reviewer: L70: I fully acknowledge that the authors use a wavelet approach. However, the principle of wavelets should be described in some more depth, as not all readers will be familiar with the principles of the wavelet transforms described here.*

Author: We agree, and will provide further explanation.

*Reviewer: L76: Fig. 1 presents data from Experiment 1a - these data have not been described and no reference is found to an experimental study. This is confusing. For example, how does the reader know the shown profile originates from a riffle-bar sequence and that the MODWT is aligned with the thalweg elevation profile? In this context, why is the CWT not aligned with the thalweg-profile? It is also a wavelet transform that is based on the thalweg-profile - so it should be aligned with this profile? This could be formulated more clearly.*

Author: A reference and note have been provided for the data that has been used. I believe the word 'aligned' may have caused some confusion and requires clarification. Both the MODWT and CWT analyses are performed on the same profile, but the CWT analysis yields a set of wavelengths that do not resemble the original signal. We have clarified this in the revised manuscript.

*Reviewer: L84: I fully agree but would also argue that almost all gravel bed rivers are in the hydraulic rough regime. In this context, what is meant by the statement that roughness correlations have only been developed for limited ranges of Re*? Typically, three ranges are distinguished, and existing approaches can be found that cover all these ranges.*

Author: This may be a simple miscommunication - we intended to say that approaches have been developed for discrete ranges of Re*, and thus, it is important to use the roughness correlation that matches the approximate Re* of the dataset. We have re-phrased this.

*Reviewer: L85: I am not sure that the width to depth ratio is an appropriate measure to determine if the flow is 2D. For example, the flow in the roughness layer of gravel beds is far from being 2D. Note also 2D conditions also depend on the relative submergence (I acknowledge that this is mentioned in the next paragraph, but I would expect such a statement already here).*

Author: In rough-bed flows, the flow is highly heterogeneous within the so-called roughness layer which extends for 2-5 roughness length scales within the flow depth (e.g., Nikora et al. 2001 Spatially averaged open-channel flow over rough bed) and it is homogeneous at higher elevations (e.g., Nezu and Nakagawa 1993 "Turbulence in open-channel flows"). A rough-bed flow is 2D far from the bed if the flow depth is large compared to the roughness length scale, in this case, the ratio channel width to flow depth is sufficient. For low submergence flows the situation is more complex and the answer is not trivial. Cameron et al. 2017 "Very-large-scale motions in rough-bed open-channel flow" observed that turbulence

statistics in flows over a homogeneous rough bed preserve two-dimensional distributions for relative submergences as low as 2. We agree with the Reviewer that a large width to flow depth ratio is generally not sufficient to guarantee two-dimensional flow conditions for flows with flow depth comparable to the roughness length scale. We, therefore, acknowledge that our flows characterized by the lowest relative submergence may not be two-dimensional, and clarify that although width-depth ratio is an indicator of 2D flow, it does not guarantee it.

*Reviewer: L100: How was discharge measured? L101: How was sediment transport scaled?*

Author: Discharge was set and monitored using a high-precision electronic pump, which would generally hold a flow rate to within pm 0.02 L/s (we do not have access to specifications at this moment due to COVID-19 lab closures). Sediment transport is not scaled within the model, and rather, it is determined by length and time-scaling. The experiment had a recirculating sediment supply, so that the sediment transport rate was determined by the system itself.

*Reviewer: L104: What was the working principle of these gages and what was the distance between them? Investigating Table 2, I found that the water depths were very low (< 0.02 m). This means that surface tension can impose a significant scale effect biasing the results. This needs to be discussed. In this context, what was the accuracy of the water depth readings? I am asking because the differences in water depth are within the millimeter range.*

Author: We agree that this could use some more clarification. We have explained that ten equally spaced gauges were used, and water depths were recorded to the nearest 1 mm. They were spaced every 1 m for all experiments except for the 8cm-wide experiments, where they were spaced every 80 cm due to the slightly shorter length of the flume, which has been detailed in the revised manuscript. With regards to surface tension, the water is dyed a rich blue colour, and the water-surface gauges were viewed almost side-on, meaning that surface tension effects could be identified and disregarded, and therefore systematic bias towards higher readings was minimised.

Based on an analysis of measurement error using the 1 mm precision, errors of between plus-minus 6 and 11 percent could be expected for mean hydraulic depths (errors are variable due to different depths), with a median of plus-minus 7.6 percent. The magnitude of error is almost the same for the velocity estimates based on propagation of error from both the discharge and water-depth measurements. We estimate that the ratio $U/U^*$ has a median error of plus-minus 11.5 percent, with a maximum of plus-minus 15 percent for the shallowest depths.

*Reviewer: L111: How exactly can discharge be scaled with the width of the experimental channel when using a Froude-scale model? The information that bankfull discharge was used is enough, but it should be stated how deep the channel was. Also, is there information available how much discharge was conveyed through the sediment bed (this could be important given the low discharges)?*

Author: The discharge was scaled to the width to maintain the same reach-averaged shear stress and the same initial relative submergence, however, this information may be unnecessary to provide within the context of this study. Due to the fixed banks, the channel has a depth that is dependent on the degree of scour/deposition and the discharge. There is no information regarding how much discharge was conveyed through the sediment bed. The grain size distribution is predominantly sand which reduces the potential for infiltration and subsurface flow. Also, before ramping up the flow to the target discharge, we monitored the water table within the stream bed (via a hole) to ensure that the bed was fully saturated. We have provided this explanation in the text.

*Reviewer: L116: Which weirs? Was there a backwater effect?*

Author: There are two weirs, one at the upstream and the downstream end, that hold the sediment within the stream table, but do not hold back the water (the initial bed is screeded to the height of these weirs). At the downstream end, where water free-falls over the weir, there is slight and localised lowering of the water surface due to a downdraw effect, but no discernable backwater.

*Reviewer: L117: What was the corresponding flow rate? That would help to answer my question @L111?*

Author: The low-flow was approximately 0.4 L/s for the 30cm channel, and 0.15 L/s for the 8cm channel. These values corresponded to the flows at which the bed could be wet and saturated without sediment transport. We have added a note to the text.

*Reviewer: L118: Had the draining of the bed any effect on the topography and was the bed saturated again when increasing the discharge after the measurements? Table 2 lacks of units and k*_s,pred has not been defined properly (see below).*

Author: The draining of the bed was rapid as the pump was simply turned off. During the initial 5-10 seconds of drainage, a small layer of sediment sourced from the riffle tails was transported into the pools, but there was no discernable change to the morphology. The bed was saturated again before increasing the discharge, as explained above. More clear units have been provided for these parameters. We have responded to comments regarding k*s,pred below.

*Reviewer: L121: Referring back to my comment @L104 - it is stated that the water surface elevation was determined to the nearest millimeter and that the mean water depths are lower than the maximum water depth (what is meant by the statement "most of the maximum flow depths"?), the degree of precision is different than stated @L122.*

Author: We have revised this section based on the previous comments and added the analysis of error. Measurement precision is 1 mm.

*Reviewer: L141: Figure 3 shows the thalweg for the total length of the table of Exp1a and the corresponding profile is shown in Fig.1a. This means to me that there is an effect of the inflow and outflow section on the bed morphology (which is directly visible in Fig. 1a) which in turn may affect the results of the wavelet analysis. This needs to be evaluated. There is also a mismatch between the thalweg elevations the figures - please explain. Moreover, the thalweg is also meandering - how was this accounted for? Note also that it was mentioned that the bricks represent a linear reference elevation. Looking at the colors of the bricks and the color-scale of Fig. 3 I would actually disagree with this statement (note also that no units are given for the scale). Another comment concerns the wetted width which, according to the numbers presented in Table 2, was different, even for bankfull conditions. I calculated the wetted width for Exp1c(3) from the numbers presented in Table2 and the formulas given at L143 and 144 and obtained a wetted width of 0.32 m which is larger than the channel width of 0.30 m. How is that possible? This in turn raises some serious questions regarding the accuracy of the experimental data which needs to be discussed in much more depth given the small scale of the experiments (see also my comment regarding the surface tension). This, together with the influence of the inlet and outlet sections indicates that more work is required to substantiate the results of the study.*

Author: We do not agree with the comment regarding the effect of the channel boundary on the wavelet transform or the broader results that are presented. The concavity of the thalweg elevation profile appears greater due to the vertical exaggeration, and the effects of the inlet/outlet are both minor and spatially localized. Thus, the removal of the edges (1 m upstream and downstream) does not change the results of the wavelet analysis, and therefore, the predicted values of ks are approximately the same (Figure 1). The edges do not matter because (1) the concavity is immediately removed by the longest wavelength of the wavelet transform, and thus, does not affect the other wavelengths, (2) this very long wavelength has the smallest contribution to the roughness length, and (3) the profile is sufficiently long and has five uninterrupted pool-riffle pairs, such that whatever influence the edges have is averaged out and does not affect the results of the analysis. Also, no water surface gauges are placed within 60 cm of the upstream or downstream ends to minimise edge effects on the hydraulics, which has been noted in the revised methods section.

[Figure]

*Figure 1. Drag size distribution when (a) entire profile is used, and (b) 1 m is removed off the end of each profile. Note the alteration of ks,pred to ks,rc in the new figure.*

We do not understand what is meant by "accounting for meandering", and do not see a mismatch between the thalweg elevations and the figures. The elevations have been extracted from the DEM directly using the estimated position of the thalweg, the process for which is described in L140-141. If there is indeed a mistake, however, we are keen to correct it.

We agree and have clarified that the bricks are not a perfectly linear reference elevation, but it is worth noting that they do not need to be. The elevation of the brick tops vary by plus-minus 4 mm in elevation across the ~ 11 m of the experiment, but the distribution of heights centers around an average elevation, which then provides the appropriate reference for detrending the model. Units are present within the DEM figure, note the "m" next to the zero, but we will add a note to the legend too.

The minor discrepancy in wetted width calculations was a result of an error in the averaging of velocity values, which has now been resolved (there is almost no change in the reach-averaged values). It is also noted that there is a slight variation in the width of the model around the 30 cm target (plus-minus a centimeter or so, as noted in the text), and back-calculated values of wetted width may vary between ~0.29 and ~0.3 cm.

*Reviewer: L149: What was the range of the discharges and what was the grain-size distribution of the bed material?*

Author: We have included the range of discharges used in Hohermuth and Weitbrecht (2018) and a summary of their grain size distribution.

*Reviewer: L157: Strictly speaking, drag is not estimated. What is estimated is k_s, a roughness length.*

Author: Agreed, and we have made corrections throughout changing "drag" to "roughness length".

*Reviewer: L159: Since skewness plays an important role, it would be good to show corresponding distributions.*

Author: We agree, and the decomposition of skewness and a brief description has been included within the new manuscript.

*Reviewer: L166: What exactly is meant by "diversity of roughness peak heights"? This remains unclear.*

Author: The diversity of roughness peak heights means the variability in the elevation of the peaks of roughness elements. If there is diversity, then the roughness elements have different heights (i.e. dune crests will be at different elevations). This explanation has been added to the text.

*Reviewer: L174: I see five peaks in the profile in Fig.1 - this number is not sufficient? (I acknowledge my comment above regarding the influence of the inflow and outflow section). Nonetheless, this needs to be discussed in more detail.*

Author: The number of pool-riffle pairs is sufficient based on the figure and discussion of edge effects provided above.

*Reviewer: L175: What was the range of delta in the present investigation?*

Author: We will clarify that the delta parameter is the vertical range of peak heights divided by the mean. Values of delta are generally over 1 and may reach as high as 4 (over the 0.35 threshold identified by Forooghi et al. 2017) for the topographic wavelengths in the present investigation. The longest 3-4 wavelengths (over 2 m) do not contain enough peaks for delta to be calculated. We will provide some statistics on delta values in the revised manuscript.

*Reviewer: L178: I do not understand Figure 7 from the previous presentation of the material.*

Author: We have removed this analysis.

*Reviewer: L179: But ks is a length scale which is different from drag. Please clarify.*

Author: As above, this has been clarified.

*Reviewer: L190: I assume that the profile was detrended for the analysis, i.e. the bed slope is not considered in this analysis? L197: What is meant by k_s,pred? How was this determined?*

Author: We have clarified the detrending process and the calculation of $k^*s,pred$ in the methods. Bed slope is not considered in the roughness correlation. No detrending is required for the TRC approach given that the wavelet transform removes the overall trend. However, when applying the roughness correlation to the profile without the transform (to obtain $k^*s,pred$), a linear detrend is first applied. We will note that 'ks,pred' refers to the predicted ks value for an individual wavelength, which is consistent with its usage elsewhere in the manuscript.

*Reviewer: L201: It is stated that topographic variation tends towards zero, but above (L191) it is stated that the effective slope is greatest at grain-scale wavelengths (this could also be seen as a measure of topographic variation at another scale). This is contradictory in my opinion. Please explain.*

Author: This may appear contradictory perhaps due to the language used. The effective slope is indeed greatest at the grain-scale wavelengths as the oscillations are tightly bunched, however, these wavelengths don't have a great deal of height variation (we used the term 'topographic variation' here, which may have confused). Effective slope is related to the aspect ratio of roughness elements rather than their vertical height. Thus, despite the pool-riffle sequence having a greater amplitude than individual grains, it has a low effective slope. We have clarified in the text, in part, by mentioning that the grain scale has a relatively small height range but the oscillations have a steep slope.

*Reviewer: L024: This figure is the same as the Form Size Distribution proposed by Nyander et al (2003)? How is it possible that it is the same? Please be more specific.*

Author: The type of graph is the same but the data is different. Nyander et al. (2003) used a wavelet transform to show how topography varies across different scales in a gravel-bed flume. This has been clarified.

*Reviewer: L205: I am confused now. Here it is stated that Equation 1 is used to predict $k_s$, which is ok. But what is then the relative value of $k_{s,pred}$ (see L 197)? I also see the need to define the concept of determining $k_s$ for different wavelengths from a hydraulic point of view in more detail taking the physics into account (and the assumptions on which the determination of $k_s$ is based: For example, were local values of velocities and slopes used? How was it ensured that uniform flow conditions prevailed? What about 2D-flow conditions at the grain scale?*
*Don't get me wrong, the presented results are certainly interesting, but this needs to be elaborated in much more depth in my opinion.*

Author: By relative values we mean, it may be more valid to make comparisons for a given river, as opposed to making comparisons across different hydraulic conditions. This will be explained more clearly. Local values of velocities and slopes were not used, and we do not assume uniform flow conditions given that the model has morphologic elements that give rise to a spatial distribution of velocity. We have addressed the comment regarding the physics of the decomposition above and would welcome further discussion as there has been no sufficient answer presented in the literature.

*Reviewer: L213: The DEM is not analyzed, but the thalweg-profile extracted from the DEM.*

Author: Agreed, change has been made.

*Reviewer: L214: The mother wavelet is only mentioned in the figure caption indicating the need to present the chosen approach in much more detail. This is rather confusing (also the choice of the other wavelets indicated in Figure 7 which remains a black box to me).*

Author: The role of the mother wavelet has been clarified in the text along with the general description of wavelets above.

*Reviewer: L218: I would argue that I only see one extreme outlier.*

Author: Agreed. There is one extreme outlier, which is significant.

*Reviewer: L224: I basically agree, and this would be one step towards answering my comment @L205. However, I am not sure that I understand the statement regarding the proportionality - it should give the same value*

Author: The comparison of $k*_{s,pred}$ and the sum of (summed)$k_{s,pred}$ is an important check, as we would expect them to have different values. The two values differ as the process of signal decomposition and recomposition is characterised by wave interference. For example, for each thalweg elevation profile there are two estimates of amplitude (1) the standard deviation of elevations, and (2) the sum of the standard deviation for each wavelength decomposed using the wavelet transform. Decomposing and recombining wavelengths alters the position and magnitude of peaks and troughs in the wavelengths, and therefore, their amplitude. Wave interference may potentially confound estimates of ks using the wavelet transform. Therefore, it is important to check that $k*_{s,pred}$ and the sum of (summed)$k_{s,pred}$ are at least positively correlated. We have added an explanation to the text.

*Reviewer: L226: I still do not understand exactly how $k*_{s,pred}$ was obtained. What is meant without the transform? Does that mean the overall profile was used? This needs to be described in much more depth.*

Author: This is correct, we can clarify: "...obtained by applying the roughness correlation to the elevation profile without the transform (i.e. using the raw thalweg elevation profile)." The explanation has been restructured and added to the methods instead.

*Reviewer: L229: Why does that validate the TRC approach? The numbers deviate (see my comment @L224). A comment in between - all this compares (to my understanding) the presents results in regard to the approach of Forooghi et al. (2017) - but how does that approach relate to the real k_s value? That means what is the "real" k_s value from the experiments? This needs to be discussed in depth. It seems that the hydraulic data have not been used to determine k_s (I might be wrong here, but this indicates that a more precise presentation of the material is required).*

Author: The reviewer is correct; we did not initially calculate the ks value using the hydraulic data. Upon this recommendation, we calculated the ks value using the hydraulic data and the rough-flow form of the Colebrook-White equation, using coefficients for open-channel flow from Keulegan (1938). We then compared this estimate of ks ($k^*_{s,CW}$) to the roughness correlation estimate (now termed [$k^*_{s,rc}$] rather than [$k^*_{s,pred}$]) (Figure 2). The experiments conducted for this study (PBR pool-bar-riffle, and PB plane-bed) have values of $k^*_{s,rc}$ that are consistently within a factor-of-two of the Colebrook-White ks values.

In the case of the step-pool experiments, there is a significant under-prediction of ks by the roughness correlation of around an order-of-magnitude, which may be explained by the effect of lower relative submergence (median h/d84 = 1.48, rather than 3.91). We believe that the similarity of the two estimates of ks for the lower gradient experiments may alleviate the concerns of the reviewer regarding the assumptions of the TRC approach (use of a single profile, ignoring slope, etc.), and also helps to position estimates of ks within a familiar framework for readers.

[Figure]

*Figure 2. Comparison of ks calculated using the Colebrook-White formula and the roughness correlation. Data has been categorized based on the morphology: PBR = pool-bar-riffle (30cm and 45cm experiments), PB = plane-bed (8cm experiments), SP = step-pool (Hohermuth experiments).*

*Reviewer: Figure 9: First, see my comments regarding the experimental data. Second, why is $k^*_{s,pred}$ used here and not sum(k_s,pred). This is confusing, as the latter parameter has been derived but is not presented in this final plot.*

Author: I believe this will be, in part, clarified by the above explanation. k*s,pred is used rather than sum(ks,pred) because wave interference affects the value of ks,pred. In short, if one is interested in the decomposition of the roughness length into different scales, then the TRC approach may be used (i.e. ks,pred for each wavelength). However, if the total value of ks is of interest, k*s,pred should be used, which is why this value was used for Figure 9 (the prediction of total flow resistance).

*Reviewer: L251 and following: This main information here should have been presented in the introduction in my opinion.*

Author: There may be an argument for this structure. However, we suggest that the implications for geomorphology are more appropriately located at the end. Based on comments from another reviewer, we have strengthened this discussion and added a range of potential applications.

*Reviewer: L268 and following: Please consider my above comments. References: The short communication is overloaded with references*

Author: We have removed several references from the manuscript.

---

## Author Comment (AC2) · 27 Aug 2020

**General comments**

We would like to thank the reviewer for suggesting where communication could be enhanced, and for encouraging us to consider the broader implications and applications of our research. First, general comments are presented and then followed by specific comments we feel appropriate to address at this time.

*Reviewer: In the present study, only one single longitudinal profile is analysed. Although it is mentioned that "multiple streamlines (parallel or even intersecting) could be employed", I think that this aspect should be discussed in more detail. For example, how much would the results be affected by selecting different streamlines? How representative is a single streamline for the flow conditions averaged over the cross-section?*

Author: There are two parts to this comment: (1) highlighting the limitations of the TRC approach in three-dimensional channel beds, and (2) the sensitivity of the TRC results to the position of the streamline.

We agree that a single streamline cannot be fully representative of the entire cross-section because it does not consider other resistance elements (e.g. banks, emergent bars), nor three-dimensional interactions between flow and channel topography. We suggest that, especially in simplified channels such as the ones of interest in this investigation, the thalweg elevation profile may still capture the important interactions between the channel topography and hydraulics. We would like to provide some evidence to demonstrate this.

Upon recommendation by another reviewer, we calculated the ks value using the hydraulic data and the rough-flow form of the Colebrook-White equation (which we can treat as a 'measured' ks value), using coefficients for open-channel flow from Keulegan (1938). We then compared this estimate of ks ($k*_{s,CW}$) to the roughness correlation estimate (now termed [$k*_{s,rc}$] rather than [$k*_{s,pred}$]) (Figure 1). The experiments conducted for this study (PBR pool-bar-riffle, and PB plane-bed) have values of $k*_{s,rc}$ that are consistently within a factor-of-two of the Colebrook-White ks values, centering around the 1:1 line. On the other hand, for the published step-pool data, there is significant under-prediction of ks by the roughness correlation of around an order-of-magnitude, which may be explained by the lower relative submergence (median $h/d_{84}$ = 1.5, rather than 3.9).

The relatively close relationship between the two independent estimates of k (estimates from either topography or hydraulics), suggests that for the experiments we conducted, the thalweg elevation profile is representative of in-channel processes. We have included this analysis and discussion in the revised manuscript.

[Figure]

*Figure 1. Comparison of ks calculated using the Colebrook-White formula and the roughness correlation. Data has been categorized based on the morphology: PBR = pool-bar-riffle (30cm expriments), PB = plane-bed (8cm experiments), SP = step-pool (Hohermuth experiments). Note the alteration of ks,pred to ks,rc in the new figure.*

Now we will discuss the sensitivity of results to the streamline position. When we were testing the analyses, we initially used a series of parallel profiles positioned from one side of the wetted area to the other. There was a significant difference in results (FSD and DSD) between the profiles, given that some were aligned with the thalweg and the pool-riffle undulations, and others captured a set of different features such as emergent bars. For the profiles positioned near the centerline, the only discernible variation in was in the amplitude of the large-scale variations, due to some profiles intersecting with the deepest parts of pools. Moreover, for these near-thalweg profiles, the estimated values of ks showed almost the same pattern (e.g. the proportion of grain- and form-scale ks values, general drag size distribution DSD shape). It was clear that the general results were the same if the profile intersected with the deep part of the channel cross-section.

We have included a figure showing the profiles used for Experiment 1a (Figure 2), which have been extracted using the technique detailed in the methods. There is some variation between the profiles, particularly in the depth of the pools, which may represent an actual change in morphology, or it may be due to the elevation profiles taking slightly different paths. There are even a couple of cases where the profile intersects with a bar at the very downstream end, which is an error that occurs in the profile extraction process when the topography is more complicated. However, despite these variations in profile shape, the DSD remains relatively similar once the pool-bar-riffle sequence has been formed (Figure 3a). The only changes in the ks decomposition are at the largest spatial scales, but these contribute almost no ks and are therefore insignificant. Even if we remove the final 1m of either end of the profiles (thus removing edge effects and potential errors), the DSD is still the same (Figure 3b).

In summary, the decomposition of ks is not very sensitive to the precise position or shape of the profile. It is for this reason that we are currently exploring the DSD as an index of channel character, given that different broad types of channel morphology (pool-riffle, plane-bed, step-pool, dune-ripple, etc.) seem to manifest as distinctive distributions of ks as a function of scale.

[Figure]

*Figure 2. Thalweg elevation profiles (based on estimated position) throughout Experiment 1a. Zero represents the mean elevation of the screeded bed.*

[Figure]

*Figure 3. Drag size distribution when (a) entire profile is used, and (b) 1 m is removed off the end of each profile.*

*Reviewer: Frequent reference is made to the study of Forooghi et al. (2017), but for readers unfamiliar with this study it is not always clear what is meant exactly or what meaning, for example, the term "effective slope" had in this cited study (see also my specific comment below to L160-178). Therefore I suggest providing some more information on this important background study.*

Author: We agree that more information regarding the initial study and effective slope parameter may be important. For example, we will explain in more detail the development of the roughness correlation in Forooghi et al. (2017), such as the general approach (i.e. relating different metrics of surface geometry to the total ks), and the surfaces that were used (e.g. their general characteristics). The effective slope is potentially a source of confusion given that surfaces with a high

effective slope (i.e. steep roughness elements) do not necessarily have a large range of elevations. We have taken steps to explain this by improving word-choice and using examples.

*Reviewer: The application of the TRC approach to the two different sets of flume experiments as illustrated in Figure 9 appears to result in a somewhat better flow resistance prediction than more traditional approaches (reduced root-mean-square error when using the k\*_s,pred roughness measure as compared to using the sigma_z measure), if the measure k\*_s,pred really refers to the application of the new TRC approach. However, it is not clear how k\*_s,pred was calculated, this needs to be clarified.*

Author: The k\*s,pred parameter is calculated by applying the roughness correlation in the way Forooghi et al. (2017) originally intended, that is, the relation is applied directly to the profile without any use of the wavelet transform. We have clarified this in more detail in the methods section of the revised manuscript.

*Reviewer:  The presence of large wood on the streambed can substantially alter the total resistance, and that particularly in such a situation using sigma_z as compared to using e.g. a characteristic grain size such as D84 improves flow resistance calculations. Can you speculate if and how using the TRC approach could further improve flow resistance calculations in such settings?*

Author: We agree that this is an important consideration. Given that the TRC approach uses surface geometry alone, any features on the channel bed surface (live or dead vegetation, large immobile grains, human structures) are incorporated within the estimates of ks. Isolating the role of large wood could be achieved with some creative thinking. For example, using structure-from-motion datasets of mountain channels, one could classify areas of large wood, remove them, and then re-interpolate to create a seamless bed without any wood. The TRC analysis could be applied to both the original DEM and the wood-less DEM (using perhaps a set of parallel profiles). The difference in ks could be attributed to the influence of large wood.

One could also perform the reverse by adding roughness elements to a profile to estimate the corresponding increase in flow resistance. This technique has applications in engineering (stabilization, flood-risk analysis) and re-naturalisation (geomorphic and biological), of rivers. We will be providing a brief discussion of these potential applications in the revised manuscript.

Reviewer:  I see one potentially interesting further application of the TRC approach with regard to the question of stress partitioning in gravel-bed streams which is one important approach to improve bedload transport predictions in these channels (e.g. Ancey, 2020, JHR, part 2, https://doi.org/10.1080/00221686.2019.1702595). Some discussion of this aspect would be welcome.

Author: We are familiar with the concept of shear-stress partitioning but have yet to directly consider the application of the TRC to this problem. From our understanding, stress-partitioning is necessary to make accurate predictions of sediment transport because only grain drag acts to move sediment, and form drag dominates the momentum budget. A common approach to determining form drag is to subtract grain drag from the total stress, where grain drag is calculated using empirical relations (usually sourced from flume experiments with flat, planar beds, and using some representative roughness value). The TRC approach provides a more direct means of partitioning drag across different scales. However, one of the key limitations here is that the TRC approach, in its current form, does not incorporate the effects of slope or relative roughness (it assumes a flat surface as well as fully-developed flow), which are especially important in bedload transport processes. We will include a discussion of this topic in the revised manuscript.

**Specific comments**

*Reviewer: Figure 1b): indicate which line refers to grain and form wavelength.*

Author: We will now use a dashed line for the form wavelength.

*L160-178: In the process of selecting an appropriate correlation between roughness measures and elements of the wavelet analysis, the authors refer to the study of Forooghi et al. (2017) who used a variable called "delta" (a measure of the diversity of roughness peak heights), and report that "effective slope is approximately proportional to drag in the range 0 < ES < 0.35". It is not clear whether the authors also determined "delta" or not. Furthermore, in eq. (2) a critical value of delta = 0.35 is used to separate the two ranges, whereas later in the text a (critical) value of 0.35 is associated with ES. (L172). This is all somewhat confusing and requires clarification.*

Author: We agree that this is important to explain. We will first clarify that the delta parameter is the vertical range of peak heights divided by the mean. By identifying peaks in each wavelength, we found that delta was generally over 1 for our experiments and was as high as 4 (over the 0.35 threshold identified by Forooghi et al. 2017). The longest 3-4 wavelengths do not contain enough peaks for delta to be calculated. We will provide some statistics on delta values in the revised manuscript.

A critical value of ES has been identified to be 0.35, which separates two regimes, termed 'waviness' and 'roughness' (Schultz & Flack, 2009, "Turbulent boundary layers on a systematically varied rough wall"). In the waviness regime, where ES < 0.35, there is a positive correlation between ES and the roughness length (that is if the height range of the roughness remains the same and the aspect ratio changes). In the roughness regime, ES has far less effect on the roughness length. The 0.35 thresholds for both ES and Delta appear to be a coincidence. This can be clarified in the manuscript.

*Reviewer: Figure 4: If the vertical dashed line is meant to indicate Dmax, it should plot at 0.008 m (L102).*

Author: We picked this up after submission and have corrected.